# Stabilizing polymer electrolytes in high-voltage lithium batteries

Snehashis Choudhury [1,11], Zhengyuan Tu [2,11], A. Nijamudheen [3,4,5,6,7,11], Michael J. Zachman [8,10], Sanjuna Stalin[1], Yue Deng[1], Qing Zhao [1], Duylinh Vu[1], Lena F. Kourkoutis [8,9], Jose L. Mendoza-Cortes [3,4,5,6,7] & Lynden A. Archer [1]

Electrochemical cells that utilize lithium and sodium anodes are under active study for their potential to enable high-energy batteries. Liquid and solid polymer electrolytes based on ether chemistry are among the most promising choices for rechargeable lithium and sodium batteries. However, uncontrolled anionic polymerization of these electrolytes at low anode potentials and oxidative degradation at working potentials of the most interesting cathode chemistries have led to a quite concession in the field that solid-state or flexible batteries based on polymer electrolytes can only be achieved in cells based on low- or moderate-voltage cathodes. Here, we show that cationic chain transfer agents can prevent degradation of ether electrolytes by arresting uncontrolled polymer growth at the anode. We also report that cathode electrolyte interphases composed of preformed anionic polymers and supra-molecules provide a fundamental strategy for extending the high voltage stability of ether-based electrolytes to potentials well above conventionally accepted limits.

[1] School of Chemical and Biomolecular Engineering, Cornell University, Ithaca, NY 14853, USA. [2] Department of Material Science and Engineering, Cornell University, Ithaca, NY 14853, USA. [3] Department of Chemical & Biomedical Engineering, Florida A&M–Florida State University, Joint College of Engineering, 2525 Pottsdamer Street, Tallahassee, FL 32310, USA. [4] Materials Science and Engineering, High Performance Materials Institute, Florida State University, 2005 Levy Avenue, Tallahassee, FL 32310, USA. [5] Department of Scientific Computing, Florida State University, 110 North Woodward Avenue, Tallahassee, FL 32304, USA. [6] Condensed Matter Theory, National High Magnetic Field Laboratory (NHMFL), Florida State University, 1800 East Paul Dirac Drive, Tallahassee, FL 32310, USA. [7] Department of Physics, Florida State University, 77 Chieftan Way, Tallahassee, FL 32306, USA. [8] School of Applied and Engineering Physics, Cornell University, Ithaca, NY 14853, USA. [9] Kavli Institute at Cornell for Nanoscale Science, Cornell University, Ithaca, NY 14853, USA. [10] Present address: Center for Nanophase Materials Sciences, Oak Ridge National Laboratory, Oak Ridge, TN 37831, USA. [11] These authors contributed equally: Snehashis Choudhury, Zhengyuan Tu, A. Nijamudheen. Correspondence and requests for materials should be addressed to J.L.M.-C. (email: mendoza@eng.famu.fsu.edu) or to L.A.A. (email: laa25@cornell.edu)

Small-molecule linear and cyclic ethers ("glymes") and their carbonate esters formed by reaction with carbon dioxide have emerged as the most important family of electrolytes for lithium and sodium batteries. These molecules are attractive for a variety of reasons, including their low viscosity and ability to coordinate with alkali metal ions, producing higher concentrations of mobile charge carriers than one would anticipate from classical theory based on their dielectric constants alone[1–6]. Macromolecular analogs, most notably polyethylene glycol dimethyl ether (PEGDME), have been reported to offer additional beneficial effects, including orders of magnitude higher mechanical modulus, low volatility, and low flammability, making them attractive candidates for solid-state or flexible lithium batteries in a variety of form factors[7–11]. A substantial body of work focused on charge carrier transport mechanisms in polyethers has shown that alkali metal ion mobility is coordinated with molecular motions and that charge carrier transport occurs predominantly in the amorphous phase of the materials where molecular mobility is highest[12–16]. A less studied, but an important trait of ethers is the ease with which they can be electropolymerized at the reducing potentials at a lithium or sodium metal anode, as well as the ease with which they degrade at the oxidizing potentials of the cathode. Almost nothing is known about how these processes can be regulated to produce self-limiting interphases and how fast ion transport at such interphases might be used to stabilize deposition processes at the anode.

Reduction of small-molecule ethers and carbonate esters has been investigated at the lithium anode and is now known to produce less mobile polymeric species by ring-opening and/or anionic chemistries[2,4,17–19]. In favorable situations (e.g., at the graphitic carbon anode of state-of-the-art lithium ion batteries) the reactions are self-limiting and produce a thin coating of a low-molar mass polymer-rich phase (interphase) at the electrode surface. This so-called solid-electrolyte interphase (SEI) limits molecular access to the electrode surface and prevents continuous electrochemical breakdown of electrolyte components. A well formed SEI is therefore crucial for stable, long-term battery operation, but almost nothing is known about how the tools of polymer chemistry can be used to harness it to achieve a similar electrochemical function at more unstable (chemical and morphological) alkali metal anodes. In cells that use lithium metal as anode, spontaneously formed interphases are in fact rarely self-limiting. Numerous studies have consequently begun to appear that center on materials synthesis strategies for creation of specially designed self-limiting interfaces on such anodes using sacrificial, easily reduced species added to an electrolyte[20–23], or application of ion permeable coatings formed ex-situ[24–26].

At the intercalating composite cathodes (e.g., NMC, LMO, LCO) of greatest contemporary interest for lithium batteries, electrolyte–electrode interfaces are not restricted to planes. Designing self-limiting interphases able to reduce/prevent electrolyte oxidation is therefore far more complex. Because ethers are particularly vulnerable to oxidative attack, a concession in the field is that ether- and polyether-based electrolytes generally cannot be used in practical electrochemical cells that employ high-voltage cathodes[27]. As a consequence, solid-state ceramic electrolytes have emerged in recent years as the most promising candidates for all solid-state lithium batteries.

Here, we reconsider the chemical processes responsible for uncontrolled interphase polymer chain growth at the anode and oxidative degradation of ethers at the cathode of a high-voltage lithium cell and on that basis show that with careful attention to designing interphases that limit polymerization at the anode and which promote de-solvation at the cathode, electrolytes based on ethers can be designed to overcome conventional limitations. We show in particular that inhibition of anionic polymerization of electrolytes using chain transfer agents (CTAs) offers unusually high levels of interphase stability at a lithium metal anode. We further report that conductive coatings of anionic molecules that can desolvate Li$^+$ ions on their way to the cathode are an integral component in designing self-limiting cathode electrolyte interfaces (CEIs) able to stabilize glymes at highly oxidizing electrode potentials. Taken together with recent work showing that polyethers in a variety of cross-linked configurations are able to inhibit rough, dendritic electrodeposition at a lithium metal anode during battery recharge, the results reported herein provide a path towards safe, cost-effective solid-state and flexible batteries based on polymeric electrolytes[7,28,29].

## Results

**Stability of polyether electrolytes at the Li anode.** We study an electrolyte comprised of bis(2-methoxyethyl) ether (diglyme) and lithium nitrate (LiNO$_3$) salt. The choice of LiNO$_3$ as the primary salt for our studies is based on the fact that it is a lower cost alternative to other salts (e.g., LiTFSI and LiFSI) reported to form electrochemically stable decomposition products in the presence of metallic Li[30]. Diglyme is chosen as the simplest oligo-ether that offers the combination of a high boiling point (162 °C) and appreciable ion transport rate at ambient temperature to be of interest as an electrolyte for a room temperature lithium metal battery. The chemical structure of the electrolyte, including the ease with which the molecule can be electropolymerized at the cathode or anode of an electrochemical cell is shared with all ether-based liquid and solid polymer electrolytes, which means that the interfacial polymerization, oxidative breakdown, and transport characteristics of diglyme at electrodes are to a reasonable approximation representative of a much broader class of polymer electrolyte candidates. Later we show that the findings using diglyme can be readily applied to achieve stable cycling of high-polymer ethers at potentials previously considered outside their stability limits.

The salt concentration in an electrolyte is a key control variable for regulating its conductivity and viscosity. Here we systematically adjusted the concentration of LiNO$_3$ to vary the ratio, $r$, of Li$^+$ cations to ether oxygen (EO) molecules in the electrolyte. Supplementary Fig. 1a reports the effect of $r$ on the temperature-dependent electrolyte conductivity. The conductivity values at room temperature are seen to exceed 1 mS cm$^{-1}$ for all materials used in the study, but there are appreciable variations at sub-zero temperatures. It is clear from the results that diglyme–LiNO$_3$ electrolytes with $r = 0.1$ exhibit the highest conductivity across the range of measurement temperatures employed in the study. It is also notable that even at a temperature of −30 °C, the conductivity of this electrolyte is >1 mS/cm, which makes it suitable for low-temperature battery operation without any compromises in power density. The continuous lines in Supplementary Fig. 1a shows that the Vogel–Fulcher–Tammann (VFT) model, $\sigma = A \exp(-E_a/R(T - T_o))$ provides a reasonable description of the measured conductivities over a rather broad temperature range. Here, $E_a$ is an apparent activation energy and is related to the free volume required for Li$^+$ ions to transport through the electrolyte; $T_0$ is related to the glass transition temperature, $T_g$, of the polymer (typically found to be on the order of $T_g$—50 °C). $E_a$, obtained from this analysis therefore provides a measure of the facility with which ions move in an electrolyte and is reported as a function of $r$ in Supplementary Fig. 1b. $E_a$ is seen to increase monotonically with $r$, similar to the glass transition temperature ($T_g$), also plotted in Supplementary Fig. 1b. This result indicates that there are high levels of molecular association between the diglyme and the salt,

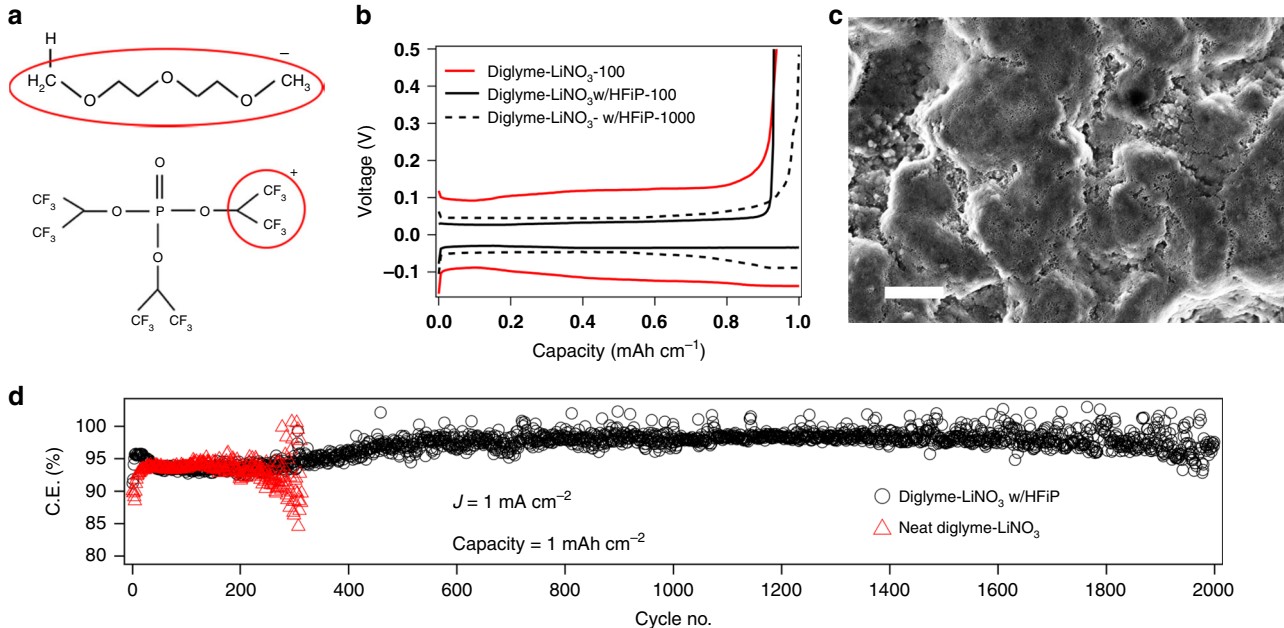

**Fig. 1** Enabling stable electrodeposition of lithium metal. **a** Schematic showing the possible cleavage sites for diglyme and HFiP molecules such that the uncontrolled polymerization of diglyme is quenched by the $CH(CF_3)_2^+$ radical. **b** Voltage profile for the electroplating and stripping of lithium metal at the same current density. The different numbers represent the cycle index. **c** Scanning electron microscopy image of stainless-steel substrate after lithium deposition for 6 h at a current density of $1\,mA\,cm^{-2}$, the scale bar corresponds to $2\,\mu m$. **d** Coulombic efficiency measurements in a Li||stainless steel asymmetric cell at a current density of $1\,mA\,cm^{-2}$ and capacity of $1\,mAh\,cm^{-2}$. The black circles represent the diglyme–$LiNO_3$ electrolyte with the HFiP additive and red triangles are results for the control/neat electrolyte

consistent with the idea that as the salt concentration is increased, diglyme molecules move in a more coupled manner. On the basis of these results, we utilize an electrolyte with $r = 0.1$ for all subsequent studies.

Glyme or ether-based electrolytes are known to undergo anionic polymerization at the surface of alkali metals, particularly at the high reducing potentials at the working anode. The resultant polymer-rich interphases are desirable because they passivate the electrode against parasitic chemical reactions with the electrolyte. Glymes are for this reason among the most preferred electrolytes for electrochemical cells in which alkali metals are to be used as anodes[1,19,30]. Unfortunately, left unchecked, the polymers formed may grow to such high molecular weights that $Li^+$ transport to the electrode is severely retarded. Alkali metals are thought to drive this process by initiating polymerization by cleaving a proton from the side-chain of a glyme molecule as shown in Fig. 1a. The polymer chain grows by an addition process wherein the active anionic reactive center collides with another glyme molecule, extending the length of the chain. Because electrostatic interactions between active centers prevent collisions between growing chains and centers can be stabilized by Li ions in solution, the growth can in principle progress indefinitely to produce extremely large, poorly con- ductive polymer chains or until all available glyme molecules are integrated into the growing center. In either event, ion mobility in the electrolyte bulk falls and interfacial resistance rises, producing premature failure of the cell by a process termed "voltage run- away".

We hypothesize that an electrolyte that addresses this fundamental, termination-free characteristic of anionic addition polymerization could limit chain growth to produce self-limited SEI on a metallic electrode. To test this idea, we employ the molecule tris(hexafluoro-iso-propyl)phosphate (HFiP) that is known to readily form multiple $CH(CF_3)_2^+$ species per molecule[31,32]. The large number of electron withdrawing groups

near the cationic fragments should enable rapid, and efficient quenching of anionic polymerization of glyme molecules by the chain transfer mechanism depicted in Fig. 1a. As a proof of concept, we performed a simple analysis wherein lithium metal was dipped in diglyme–$LiNO_3$ electrolyte, with and without the HFiP molecule, and aged for 1 month. Supplementary Figure 2 shows that in the absence of the CTA the electrolyte turns yellow due to uncontrolled bulk polymerization of the diglyme molecules, while the lithium is blackened due to surface reactions. In comparison, electrolytes that contain the HFiP molecules not only the diglyme solution but also the lithium surface maintains its pristine form. To interrogate the interphases formed on the Li specimen, the lithium foil employed in both experiments was analyzed using X-ray photoelectron spectroscopy (XPS) and the results reported in Supplementary Figs. 3 and 4. The F-1s XPS for the case with HFiP additive manifests a single peak at 688.9 eV representing –$CF_3$ bond, which is further confirmed from the C- 1s XPS from the peak at 293.3 eV, while it is absent in the C-1s for the lithium extracted from HFiP-free electrolyte[33]. The absence of a metal-fluoride binding energy peak provides confirmation that the –$CF_3$ groups do not decompose in the presence of the lithium metal electrode, ruling out an alternative stabilizing mechanism reported in our previous work[22,23].

The effectiveness of the approach to create self-limited interphases on a Li anode was evaluated in an asymmetric electrochemical cell comprised of lithium metal and stainless-steel electrodes. By comparing the electric current generated when a specific amount of Li is stripped from the Li electrode and deposited onto the stainless-steel electrode, with the current required for the reverse process, the coulombic efficiency (CE) of the cell can be determined. Figure 1b shows the voltage profile during a typical measurement in cells with and without the HFiP CTA. It can be seen that although for the 100th cycle the CE values for the two electrolytes are the same, the overpotential for stripping and plating Li is vastly reduced by the CTA, consistent

with expectations for the CTA's ability to terminate polymer chain growth. The consequence of these effects is quite clearly seen in Fig. 1d and Supplementary Fig. 5, which report the CE for electrolyte with and without the CTA, at current densities of 1 and 0.25 mA/cm², respectively with each half cycle comprising of 1 h. This means that approximately 5 μm and 1.25 μm of the 450 μm Li electrode is stripped and plated during each cycle, respectively. It is seen that the CE is maintained at a value >98% for 2000 plate-strip cycles, even without efforts to optimize the composition of the CTA in the electrolyte or its efficiency in terminating addition polymerization. This level of stability has to our knowledge not been observed in a lithium metal cell using any liquid electrolyte.

The benefits of the CTA are even more obvious when results for electrolytes with and without HFiP are compared (Fig. 1d). It is observed that whereas the control diglyme–LiNO₃ electrolyte with/without the CTA exhibit similar CE values for the initial 200 cycles, upon longer-term cycling large fluctuations appear in the latter that are absent in the former. Similar behavior is observed at the lower current density of 0.25 mA/cm², however the fluctuations in CE are seen after 500 cycles. We further characterize the electrodeposited Li surface using SEM analysis of the stainless-steel electrode after electrodeposition of 6 mAh cm⁻² (ca. 30 μm of Li) at 1 mA cm⁻² (Fig. 1c) and at 0.25 mA cm⁻² (Supplementary Fig. 6), in the glyme electrolyte containing HFiP. It is seen that the deposits are compact at both these current densities and that the coverage of the smooth deposits spans several microns, indicating their large-scale uniformity.

The fluctuations in CE observed in the control electrolytes are associated with the sporadic electrical connections with electrically disconnected fragments of lithium ("orphaned lithium") formed during the electrodeposition process and are indicative of the irreversibility of the process. These findings are confirmed by postmortem analysis of the electrode surface after cycling the Li||stainless steel cell with and without HFiP at current density of 1 mA cm⁻² for 100 cycles, followed by depositing lithium of capacity 1 mAh cm⁻² on the stainless-steel electrode. The SEM images of the electrodeposited stainless-steel reported in Supplementary Fig. 7 indicate that in contrast to open, dendritic or needle-like deposits as observed in the control electrolyte, the CTA containing electrolyte resulted in compact structures. We believe this difference is a reflection of faster diffusion of lithium ions and low charge transfer resistance for the anodic reaction: $Li^+ + e^- \rightarrow Li$ at interphases where polymerization of the glyme is constrained.

The continuous polymerization of the diglyme molecules without the HFiP CTA can lead to increased battery resistance in extended cycling, which can be investigated using impedance spectroscopy measurements. Supplementary Figure 8 reports Nyquist plots obtained from cells containing the diglyme–LiNO₃ electrolyte with and without CTA. The cells used in the study were comprised of lithium metal and stainless-steel discs as electrodes and were cycled 100 times at a current density of 1 mA cm⁻² and 1 mAh cm⁻² capacity, with plating in the last step. Fitting the impedance spectra with the appropriate circuit model (shown in the inset of Supplementary Fig. 8), reveals that the interfacial resistance for the cell using the control electrolyte was 77.5 Ω, while that of the CTA-containing cell was 50.9 Ω. Thus, it can be argued that the CTA enables longer term stable cycling by preventing electrolyte degradation. We also analyzed the surface of lithium metal extracted from a Li||stainless steel cell with the diglyme–LiNO₃–HFiP electrolyte after 100 cycles using XPS (reported in Supplementary Fig. 9). It can be seen that the majority of the F-1s spectra comprised of the peak at 688.9 eV corresponding to the –CF₃, it also shows the presence of a peak at

684 eV that can be attributed to the presence of LiF species. Several previous works on electrode–electrolyte interfaces have demonstrated that LiF stabilizes electrodeposition of metallic lithium[23,34]. Overall, we believe that the stabilization mechanism by quenching anionic polymerization of glymes is universal for molecular additives other than HFiP that are capable of undergoing chain transfer including, BTFE (bis(2,2,2-trifluoroethyl))[35] and metal salts based on TFSI (bis(trifluoromethanesulfonyl) imide)[36], among others.

**Anionic polymer coatings on cathode.** The success of a CTA in limiting polymer growth under the reducing potentials at the Li anode leads us to hypothesize that an analogous approach might be used to enable all ether-based electrolytes to be operated at higher potentials, where oxidative breakdown of the electrolytes is a well-known and longstanding barrier to use of ether-based electrolytes in lithium batteries that utilize high voltage cathodes. To shed light on the degradation reaction of glymes at a high voltage cathode, we experimentally characterized the products formed at the NCM cathode after cycling a Li||NCM cell using the base diglyme–LiNO₃–HFiP electrolyte at potentials between 3.0 and 4.2 V. Figure 2a reports results from Fourier transform infrared (FTIR) analysis, which show one major change relative to the spectrum of a cycled electrode in the diglyme electrolyte in comparison to the case of a Lithion-protected electrode (discussed subsequently): the occurrence of a noticeable peak at around 1600 cm⁻¹. This vibration is a well-known characteristic of unsaturated C=C bonds and we tentatively attribute it to the presence of the H-abstraction reaction illustrated in Fig. 2b. As reported further in Supplementary Fig. 15, similar C=C bond formation is also observed in electrolyte recovered from the cathode-facing surface of separators in cycled Li||NCM cells. The plausible reaction mechanism for the high voltage degradation of diglyme molecules is reported in Supplementary Fig. 22, where it is seen that aldehydes and mono-substituted alkenes are formed consistent with our experimental observations. It provides clues about how and why oxidation of ether-based electrolytes degrades battery performance: H-abstraction and formation of such groups would lead to self-polymerization of the electrolyte resulting in high overpotentials. Previous studies using [Li(glyme)₁]⁺ X-*ionic liquid* complexes have speculated that the oxidation reaction of glymes at a high-voltage lithium battery cathode involves abstraction of a lone pair from the EO at the oxidizing cathode potentials[37]. On that basis, improved levels of oxidative stability observed in the complexes was attributed to the EO donating its lone pair to the Li⁺ cation to marginally lower the highest occupied molecular orbital (HOMO) energy level of the glyme molecule[37,38]. In the present study, we further calculated the frontier molecular orbital energies and HOMO–LUMO energy gaps in diglyme and its complexes with Li and similarly observed that the HOMO energy is weakly reduced (Supplementary Fig. S23 and Supplementary Table 1).

Owing to the strong coordination of Li⁺ with glymes, it would be difficult to prevent efficient Li⁺ intercalation into the cathode without exposing the electrolyte solvent to oxidation. An approach that would desolvate the Li⁺ ions without compromising mobility at the cathode/electrolyte interface could achieve both goals. Here, we chose to study interphases formed by the semi-crystalline anionic polymer electrolyte, Lithion (see Fig. 2c). This choice is motivated by three considerations. First, we discovered that solutions of Lithion in aprotic carbonate ester and alcohols possess low viscosity, allowing the polymer to be transported by liquid carriers into the pores of preformed cathodes. Second, we previously explored electrokinetics of Lithion coated interfaces and reported that the negative charge

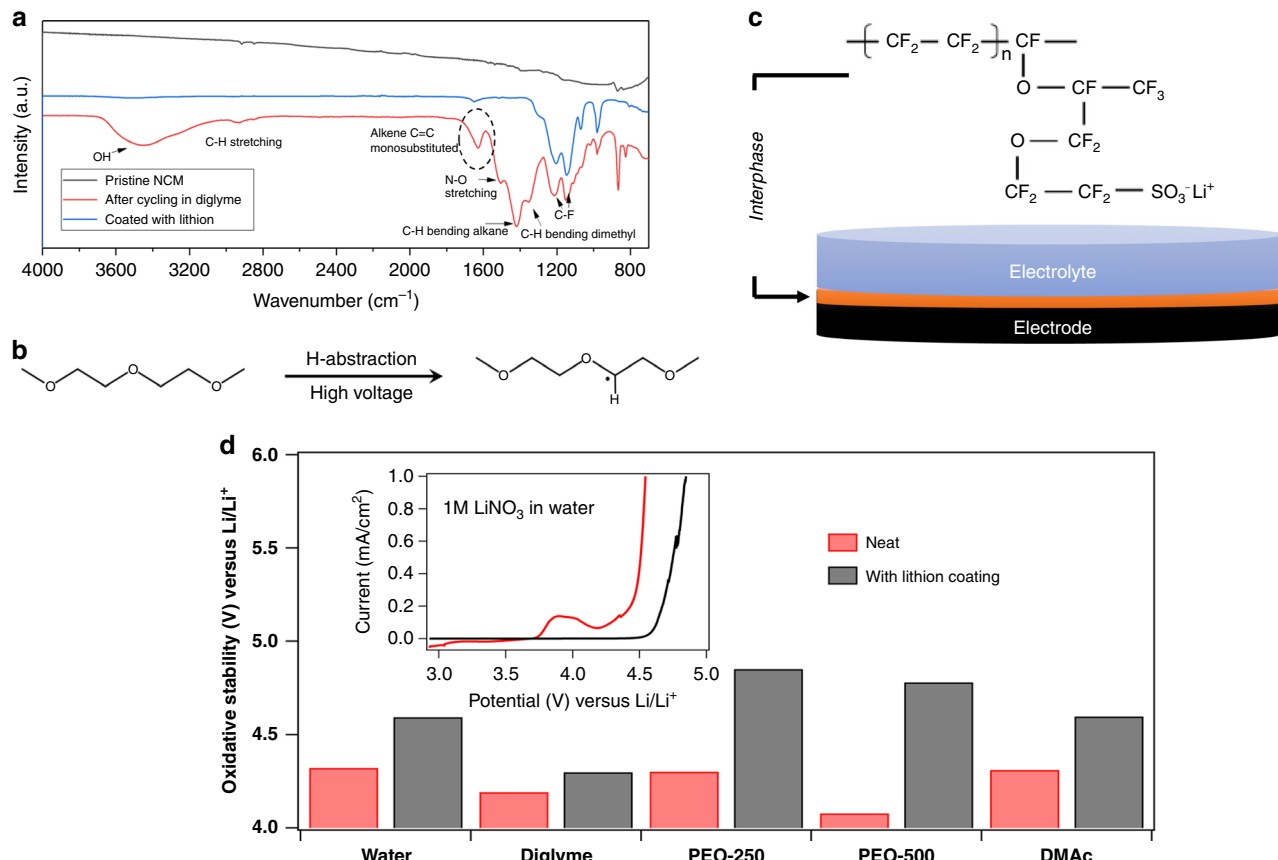

**Fig. 2** Designing stable cathode electrolyte interfaces (CEI) based on immobilized anions. **a** Intensity profile obtained from Fourier transform infrared spectroscopy (FTIR) for pristine (uncycled) NCM and NCM cathode extracted from a Li||NCM cell cycled twice at C/10, with and without the Lithion coating. **b** Schematic showing the proposed proton extraction mechanism from the diglyme molecule due to oxidation at high voltages. **c** Schematic showing the structure of lithiated Nafion$^{TM}$ (Lithion) utilized to form the artificial CEI. **d** Bar chart comparing the oxidative stability of different electrolytes with (black) and without (red) Lithion coating. The measurements were performed in 3-electrode cell with Ag/AgCl as reference and stainless steel as counter and working electrodes. The scan rate was 10 mV/s. The electrolytes investigated are 1 M LiNO$_3$ in water, $r = 0.1$ LiNO$_3$ in diglyme, $r = 0.05$ LiNO$_3$ in PEO-250, $r = 0.05$ LiNO$_3$ in PEO-500, and 1 M LiNO$_3$ in dimethylacetamide. The inset shows results from linear scan voltammetry for the 1 M LiNO$_3$(aq) electrolyte. All the voltages are shifted with respect to Li/Li$^+$

centers created by dissociation of sulfonate groups along the polymer backbone provide an effective electrostatic shield, which limits the transport of negatively charged species at planar electrodes, but allows for facile transport of cations[39]. This yielded lithium transference numbers approaching unity in liquid electrolytes that simultaneously display high bulk and interfacial ionic conductivities. Finally, the small pore sizes and coexistence of hydrophobic and hydrophilic domains in a Lithion membrane mean that at appropriate thicknesses it should be possible to retard transport of strongly polarized molecules in a solvent transport, without compromising cation mobility.

We performed linear scan voltammetry in a three-electrode cell using Ag/AgCl as the reference electrode and Lithion coated stainless steel as both the working and counter electrodes to determine the effectiveness of a Lithion layer in improving the oxidative stability of liquid electrolytes. Because the stabilizing mechanism hypothesized for the Lithion coating is general we evaluated a variety of electrolytes, including aqueous liquids to high Gutmann donor number fluids like dimethylacetamide notorious for their oxidative instability. The electrochemical oxidation potential measured in each case was compared to the case without the Lithion coating. The inset of Fig. 2d reports the oxidative window of 1 M LiNO$_3$–water as electrolyte. It is seen that the Lithion coating noticeably delays the onset of a measurable anodic current and elevates the oxidation potential

of water by at least 0.3 V. Similar voltammetry analysis for other non-aqueous electrolytes, including dimethylacetamide, diglyme, and higher molecular weight PEO are reported in Supplementary Fig. 10. Results reported in Fig. 2d show that in all of the cases an improvement in oxidative potential is observed, with the largest increases seen in electrolytes that wet the Lithion layer least. The universality of the stabilizing effect of the anionic Lithion coating, lends support to our hypothesis that the stabilization mechanism is fundamental.

Motivated by these results, we created electrochemical cells comprised of a lithium metal anode, NCM cathode coated with/ without a Lithion coating and the diglyme–LiNO$_3$–HFiP electrolyte used in the first phase of the study. A drop-cast method in which Lithion was first dissolved in isopropanol to form a dilute solution that was drop coated onto an NCM electrode disc. The thickness of the Lithion layer can be adjusted by manipulating the concentration of the cast solution. We used a scanning electron microscopy methodology (cryo-SEM) performed at cryogenic temperatures (see Supplementary Fig. 11) to characterize the thickness and distribution of the Lithion coating formed on the NCM electrode. This analysis revealed that quite thick coatings (90 μm to 30 μm) are produced by the drop casting method. Complementary analysis using EDX mapping on a cross-section milled into the layer by cryogenic focused ion beam, as shown in Fig. 3a, and across the full thickness of a cracked layer, as in

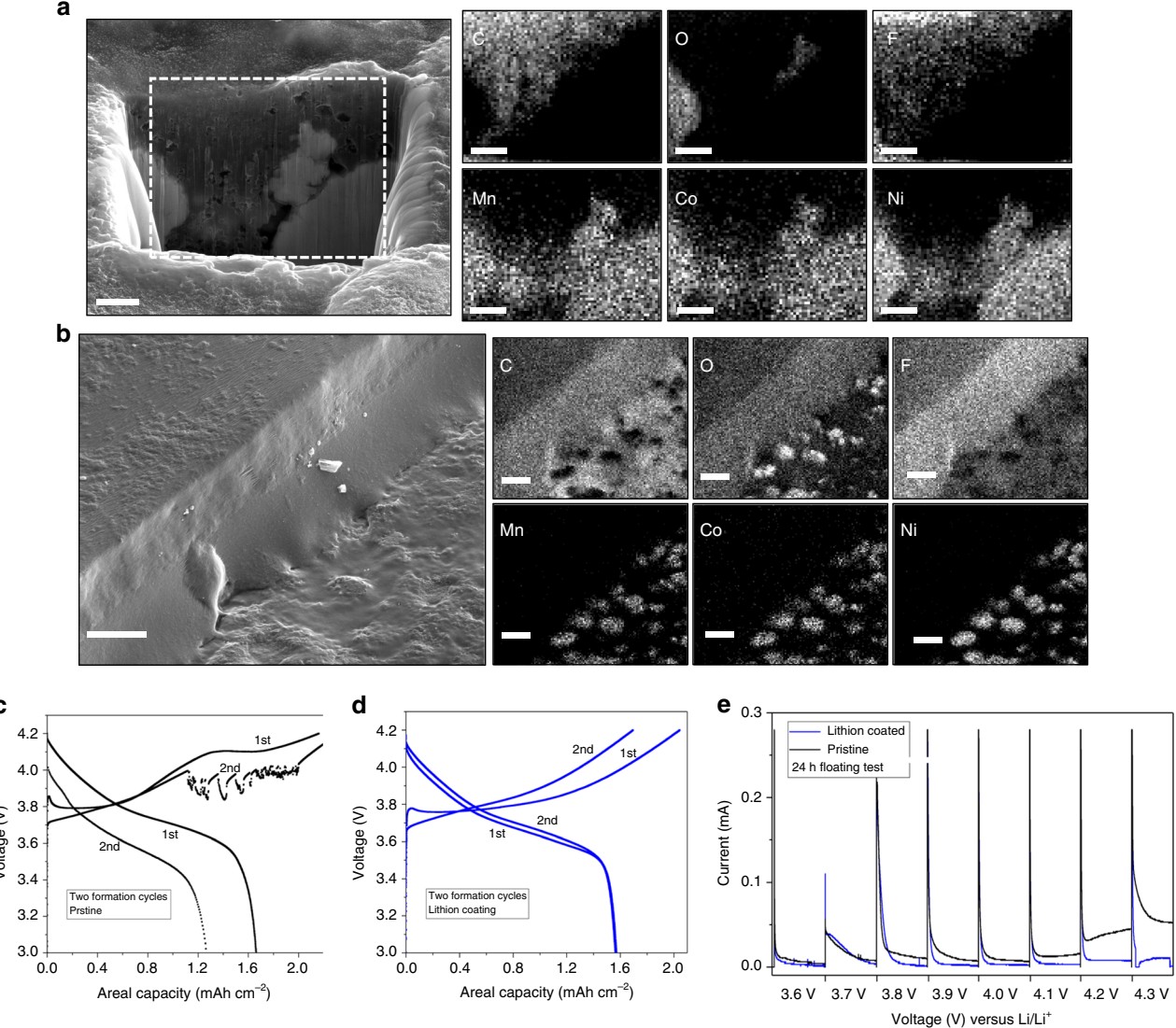

**Fig. 3** Immobilized anions at the cathode prevent glyme oxidation. **a** Cryo-SEM image of a cross-section of a Lithion-coated nickel manganese cobalt oxide (NCM) electrode obtained by focused ion beam milling. EDX mapping of different elements present in the cross-section is shown on the right. **b** Cryo-FIB/ SEM images of the Lithion coated NCM electrode surface. A Lithion layer present on the NCM cathode is deliberately cracked during preparation to reveal its thickness. **c** Voltage profile of a lithium||NCM cell using the base electrolyte of diglyme–LiNO$_3$–HFiP at C/10 rate; scale bars are 2 µm. **d** Voltage profile of Li||NCM cell using the same base electrolyte, however the cathode is coated with a layer of Lithion, operated at C/10; scale bars are 10 µm. **e** Electrochemical floating experiment in a Li||NCM cell. In these experiments, the voltage is fixed at different values ranging from 3.6 to 4.3 V for a period of 24 h and the current response measured to quantify the electrochemical stability of the electrolyte/electrode interphases over a range of potentials. The black curves represent results for uncoated NCM and blue are for Lithion-coated NCM electrodes

Supplementary Fig. 12, shows that the Lithion penetrates throughout the electrodes and engulfs the active particles (see Fig. 3b). Comparing the individual elemental maps, the C and F signals are observed around individual particles containing Ni, Mn, and Co, which suggests that the Lithion layer not only coats the macroscopic surfaces of the electrodes, but may also wrap the NCM particles.

To evaluate the effectiveness of the approach we performed galvanostatic cycling studies using Li||NCM cells containing the diglyme–LiNO$_3$–HFiP electrolyte with/without the Lithion coating applied to the cathode. The results for the baseline case (i.e., no-Lithion coating of the cathode) are reported in Fig. 3c, where it is seen that the voltage profile exhibits a prolonged charging step in the 1st cycle and erratic fluctuations in the 2nd cycle above 3.8 V vs. Li/Li$^+$. The discharge step does not show such fluctuations, however a high overpotential is observed in the

2nd cycle indicative of high battery resistance produced by oxidative degradation of glyme electrolytes. These results can be compared with observations provided in Fig. 3d for Li||NCM cells containing the same electrolyte, except where the NCM electrode was coated with the mentioned Lithion layer. It is immediately apparent from the voltage profiles for the 1st and 2nd cycles in Fig. 3d that the prolonged charging feature observed for the control cells is absent. As a more extreme test, we investigated the cycling behavior of Li||NCM cells comprised of a thin (50 µm) metallic lithium foil as anode. Results reported in Supplementary Fig. 13 show that the oxidative stability provided by the Lithion coating extends well beyond the first few charge–discharge cycles and that stable cycling is achieved for over 100 cycles. Supplementary Fig. 14 reports results from a more stringent protocol in which Li||NCM cells composed of the Lithion coated NCM cathodes were subjected to stepwise increasing charging

potentials, up to 4.4 V. Again, unprecedented levels of stability are observed consistent with the enhanced oxidative stability of the electrolytes deduced from the LSV measurements. Previous work has reported that glyme-based electrolytes that form ionic liquid complexes with Li salts can achieve stable cycling in Li||LFP and Li||LCO cells[37]. However, the level of stability achieved in Supplementary Figs. 13 and 14 with our facile cathode coating process in Li||NCM cells is unprecedented.

A more rigorous analysis of the electrolyte stability can be gleaned using electrochemical floating experiments. In this experiment, Li||NCM cells with and without the Lithion coating are charged at voltages ranging from 3.6 to 4.3 V in a step-wise ramp and the voltage maintained at a targeted value for a period of 24 h, as shown in Fig. 3e. The leakage current obtained at each voltage is recorded and can be used to directly assess the importance of electrochemical degradation of electrolytes in the fully charged state. The results show that the leak current is always higher for the control cells (i.e., without the Lithion electrode treatment) than for those that utilize a Lithion-coated NCM electrode. In addition, it is seen that the leakage currents for the pristine NCM case exceeds those of the modified NCM cells at a faster rate beyond 4 V, which is also consistent with the low CE observed in the Li||NCM half-cell cycling.

**In-situ formation of anionic species on cathode**. The effectiveness of the Lithion cathode coatings suggests that other approaches that lead to in-situ formation of anionic polymer coatings throughout the cathode would be as straightforward a strategy for enabling ether-based electrolytes in lithium cells employing high voltage cathodes. To evaluate this concept, we used quantum mechanical calculations at the level of hybrid density functional theory (DFT) to computationally study the interphases formed at high potentials by the lithium salt lithium bis(oxalate)borate (LiBOB) in a glyme electrolyte. The BOB anion is of interest because it has been reported in previous studies to readily form either an open, dianion by breaking a B–O bond, or can furnish dissociation products[40–42]. The reactions of these intermediate species with diglyme would generate distinct coupling products. We calculated the reaction free energies for the formation of a series of neutral and anionic O–C, C–C, and B–C coupling products from the diglyme and BOB dianion. These transformations proceed through the release of $CO_2$ molecules. Unique coupling products considered here, and the respective free energy changes are presented in Fig. 4a. The calculations indicate that the formation of negatively charged species is thermodynamically more favorable than the respective neutral analogs. Among the anionic dimers, the C–C coupling product (a in Fig. 4a) formed by the release of a $CO_2$ molecule is thermodynamically most favorable ($\Delta G = -0.64$ eV).

Starting from the negatively charged dimer, one could envision its subsequent reactions with diglyme and $BOB^{2-}$, which will generate oligomers, polymers, or a supramolecular assembly at the electrode–electrolyte interface. To evaluate this possibility, we calculated the reaction-free energies for the step-wise generation of neutral or negatively charged dimer, trimer, tetramer, and pentamer from $BOB^{2-}$ and diglyme (Fig. 4b). These calculations reveal that the formation of neutral or negatively charged trimer and higher aggregates is thermodynamically unfavorable. The formation of anionic and neutral forms of trimer from the dimer is endothermic by 1–4 eV, whereas the generation of higher order coupling products is highly unfavorable ($\Delta G > 10$ eV). At higher voltages, the trimers could still form, however it is very unlikely that further polymerization will occur. The higher oligomers with multiple charges may not be stable as they would readily dissociate to smaller charged dimers or trimers. On this basis we

theoretically calculate the redox potentials of the glyme molecule and its oligomers with BOB molecules, presented in Fig. 4c using the computational methodology described in the Supplementary Information. It is seen that the glyme-BOB oligomers are charged and electrochemically stable at high potentials. It can be argued that these initially generated oligomers form a network at the cathode via strong non-covalent interactions; furnishing a charged supramolecular assembly (as shown in the schematic of Fig. 5a).

To experimentally interrogate the cathode-electrolyte interphase (CEI) formed in the presence of a LiBOB salt, we cycled a Li||NCM cells with the LiBOB additive in diglyme–$LiNO_3$–HFiP (base) electrolyte twice at C/10 and extracted the NCM cathode for FTIR analysis. Figure 4d shows agreement between the measured and computationally predicted IR spectra for the oligomeric species discussed in the previous section. Differences in the relative intensities can be attributed to the presence of additional components (salt, additive, surface impurities) that are not considered in the DFT calculations. Verification that LiBOB enhances the oxidation potential was performed using linear scan voltammetry in a 3-electrode setup, as well as the more stringent electrochemical floating-point test. The results are reported in Fig. 5b and Supplementary Fig. 16, respectively. It is seen that the degradation potential of the electrolyte is enhanced by approximately 0.3 V (Fig. 5b). Since, the measurement was performed in a 3-electrode cell with stainless steel as working electrode, our argument that the LiBOB forms the CEI in-situ is strengthened. In the floating experiments with and without the LiBOB additive Li||NCM cells were charged to varying voltages and held at each voltage for 24 h. The results (Supplementary Fig. 16) show that the leakage is measurably lower than the control cells at all voltages. We also characterized the surface of the lithium metal anode using XPS after two initial cycles of charge and discharge at C/10 rate (Supplementary Fig. 17).

As a final assessment, we created Li||NCM cells comprised of a thin (50 μm) metallic lithium foil as anode with the NCM cathode with anode to cathode capacity ratio of 5:1 and the base electrolyte (diglyme–$LiNO_3$–HFiP) with LiBOB as a salt additive. The voltage profiles for the 5th, 50th, and 100th cycles are reported in Fig. 5c and the cycle life in Fig. 5d. It is seen that CE of the cells is high (>98%) and that the discharge capacity is retained to more than 80% for at least 200 cycles at a rate of C/5. Similar performance is also observed at a rate of discharge (C/2) reported in Supplementary Fig. 18. The performance improvements are consistent with those observed for the Lithion-coated cathodes.

The enabling concept of high voltage stabilization using anionic interfaces is not limited to oligomeric liquid electrolytes. It should be applicable to any PEO-based electrolyte, including gels and solid-state electrolytes. To evaluate this, we designed gel electrolytes comprising of 1 wt% of a high molecular weight PEG ($M_w = 100$ kDa) in diglyme (image shown in Supplementary Fig. 19) and investigated them in Li||NCM cells with the same designs as in the previous section, in which the LiBOB salt additive is/is not present. These cells were evaluated at a rate of C/5 as shown in Supplementary Fig. 20. Unlike control liquid electrolyte, the gel electrolyte supports stable charge–discharge profiles even without LiBOB. However, beyond 20 cycles, there is a sharp drop in the capacity followed by a noisy charge profile due to the breakdown of the electrolyte. The limited stabilization of the control gel electrolytes is likely a result of the higher viscosity of the electrolyte in the cathode pores, which limits diffusion of the C=C species created at the cathode, retarding polymerization. In contrast, it is found that the LiBOB additive in the gel electrolyte enables stable cycling and high capacity retention for at least 100 cycles. We also demonstrate that stable

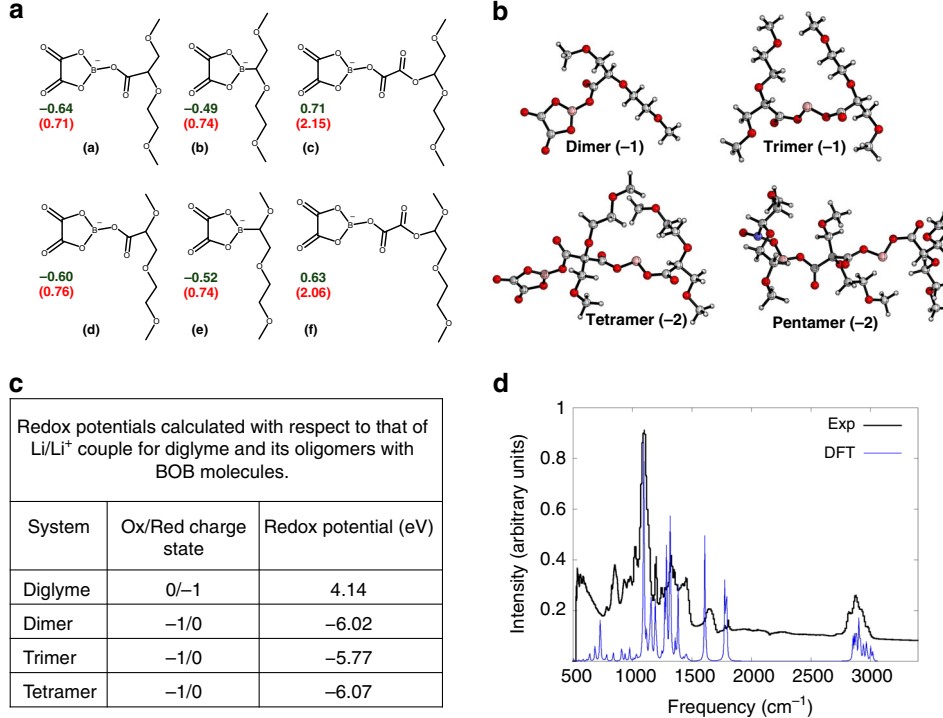

**Fig. 4** In-situ formation of anionic aggregates at cathode interface. **a** Structures of plausible coupling products of BOB$^{2-}$ and diglyme. Calculated reaction-free energies (in eV) for the formation of anionic (green) and neutral (red color) dimers. **b** Optimized geometries for the dimer and higher order coupling products of BOB and diglyme. The respective charges are shown in the parenthesis. **c** Table showing calculated redox potentials for diglyme and its oligomers with BOB molecules. Oxidation/reduction potentials are calculated with respect to that of Li/Li$^+$ couple. A positive or negative sign is used to represent reduction and oxidation potentials, respectively. **d** Infrared (IR) spectra comparing the intensity profiles obtained from experiment and DFT calculations. The experimental profile was obtained from an NCM cathode harvested from a Li||NCM cell after two charge–discharge cycles at a low rate of C/10. The cell contained a diglyme–LiNO$_3$–HFiP electrolyte with 0.4 M LiBOB as a salt additive

cycling is achieved for Li||NCM cells (see Supplementary Fig. 21) which use a recently reported cross-linked nanoparticle membrane[43,44] infused with small amounts of the base diglyme electrolyte to form a single-component solid-state hybrid electrolyte.

## Discussion

In conclusion, we have shown that cationic chain-transfer agents can be used to terminate anionic polymerization of ether-/glyme-based electrolytes at a lithium metal electrode, producing self-limiting interfaces, high CE, and extend the lifetime of the anode (to over 4000 h) in asymmetric lithium||stainless steel cells. Building on these observations, we show that a longstanding barrier to deployment of glyme electrolytes can be removed using either ex- or in-situ generated interphases in the cathode that limit transport and reduce reactivity of active polymer centers by what we hypothesize to be an electrostatic shielding mechanism. Specifically, we show that a CEI which hosts immobilized anions acts as a barrier for the oxidation reaction. The concept is extended to create in-situ generated CEIs composed of anionic polymer aggregates, which are reported to improve the lifetime of a high voltage lithium metal battery.

## Methods

**Computational details.** All structures are optimized in the gas-phase using wB97X-D[45,46] functional and 6-311G(d,p)[47] basis sets implemented in the Gaussian suite of programs[48]. Vibrational frequencies are calculated at the same level of theory to ensure that the optimized geometry represents a true minimum; i.e., no negative frequencies are found. Further, single point calculations are performed on these structures by employing a polarizable continuum model (PCM) to mimic the effects of diglyme[49]. We used a dielectric constant of 7.23 for diglyme. A value of 1.63 eV is assumed for the electron solvation free energy[50].

**Materials.** Lithium discs were obtained from MTI Corporation. Diglyme, lithium nitrate were all purchased from Sigma Aldrich. Tris(hexafluoroisopropyl) phosphate was obtained from Synquest Laboratories. Celgard 3501 separator was obtained from Celgard Inc. Lithion solution (LITHion™ dispersion, ~10 wt% in isopropanol) was purchased from Ion Power Inc. The Lithion is composed of a Nafion-type perfluorinated polymer having the sulfonic acid groups (EW ~1100) ion exchanged by lithium ions. Nickel manganese cobalt oxide (NCM) cathodes were obtained from Electrodes and More Co. All the chemicals were used as received in after rigorous drying in a ~0 ppm water level and <0.1 ppm oxygen glove box.

**Coating of NCM electrode with Lithion solution.** NCM electrodes were punched out using a hole-punch of diameter 3/8″. On a flat bench-top, the NCM cathodes were laid and ~20 μl of Lithion solution was dropped to evenly cover the entire surface. Thereafter the electrodes were dried in open air for 6 h, followed by rigorous drying in a vacuum oven at a temperature of 60 °C for 24 h.

**Synthesis of gel and crosslinked nanoparticles electrolyte.** The gel electrolyte was prepared by dissolving 1 wt% of PEG-100 kDa (Sigma Aldrich) in an electrolyte solution of diglyme–LiNO$_3$–HFiP (with and without 0.4 M LiBOB salt additive) and thereafter heating the solution to 60 °C overnight. Thereafter the gel electrolyte was brought to room temperature before usage. It was used with a 3501 Celgard separator for lithium battery cycling.

The crosslinked solid electrolyte was prepared using the same procedure reported in our earlier work[43,44]. After thoroughly drying, the membrane was soaked in the diglyme–LiNO$_3$–HFiP–LiBOB solution for a period of 2 days before using in the battery. No separator was used in these batteries.

**Dielectric spectroscopy.** The ionic conductivities of the electrolytes were measured at room temperature the desired electrolytes between two gold-plated copper discs using a Novocontrol Broadband Dielectric spectrometer with a frequency range of 10$^{-3}$–10$^6$ Hz. The electrolyte was sandwiched between the discs using a Teflon o-ring. The DC conductivities were obtained from the plateau of real part of the conductivity vs. frequency curve. The dielectric spectroscopy instrument was calibrated initially using a 1 M KCl standard solution.

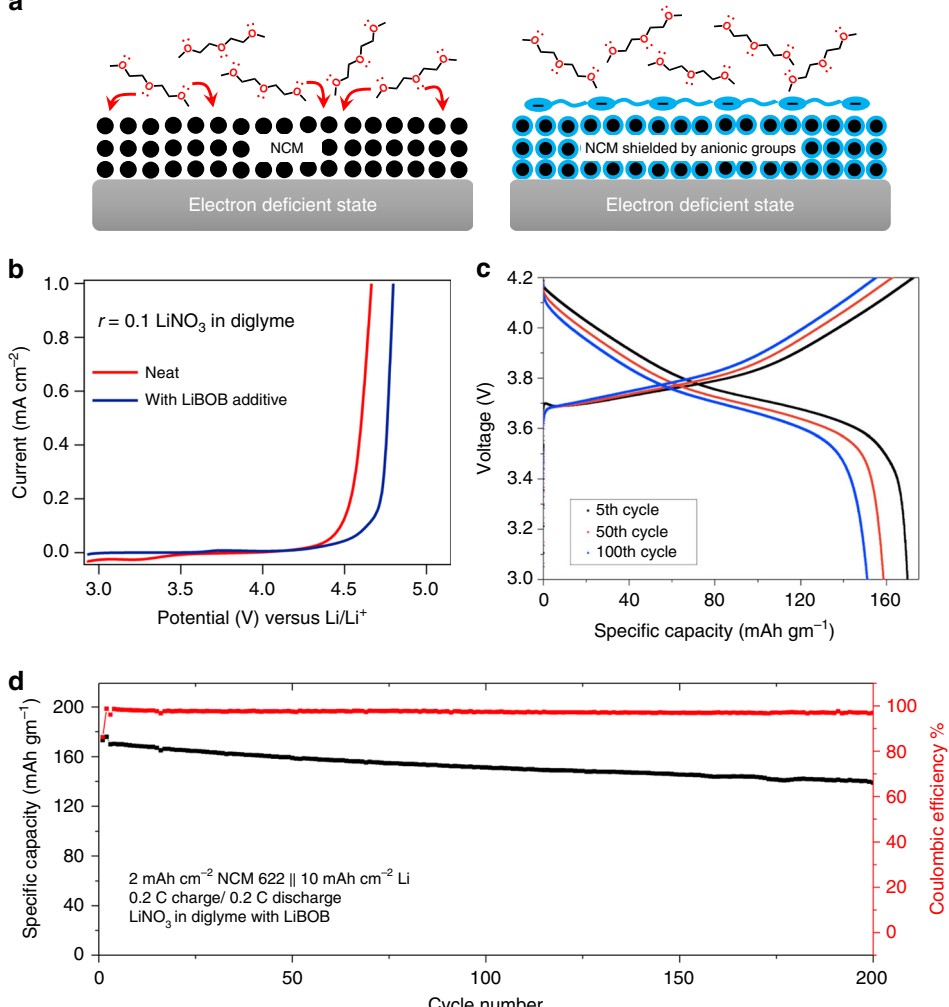

**Fig. 5** Enabling stable cycling of high voltage lithium battery with ether electrolytes. **a** Schematic showing the proposed mechanism by which oxidation of ethers is inhibited at a high-voltage CEI containing a layer of immobilized anions. **b** Potential-current diagram obtained from linear scan voltammetry in a 3-electrode cell in which Ag/AgCl is the reference electrode and stainless steel is used as both the working and counter-electrodes. The scan rate was 10 mV/s and diglyme–LiNO$_3$–HFiP, with (blue) and without (red) 0.4 M LiBOB salt additive was used as the electrolyte. **c** Voltage profile for the 5th, 50th, and 100th charge and discharge cycles of a Li||NCM cell containing diglyme–LiNO$_3$–HFiP electrolyte with 0.4 M LiBOB as salt additive. **d** Discharge capacity retention and coulombic efficiency over 200 cycles for a Li||NCM cell with diglyme–LiNO$_3$–HFiP electrolyte with 0.4 M LiBOB as salt additive. Here, a 50 μm thick Li foil is used, and the anode to cathode (N:P) capacity ratio is 5

**Scanning electron microscopy**. Surface analysis of electrodeposited stainless-steel was done using SEM with the LEO155FESEM instrument. The sample was prepared by depositing 6 mAh cm$^{-2}$ in battery comprising of lithium vs. stainless-steel comprising of diglyme–LiNO$_3$–HFiP electrolyte and Celgard separator.

**X-ray photoelectron spectroscopy**. XPS was conducted using Surface Science Instruments SSX-100 with operating pressure of ~2 × 10$^{-9}$ torr. Monochromatic Al K-α x-rays (1486.6 eV) with beam diameter of 1 mm were used. Photoelectrons were collected at an emission angle of 55°. A hemispherical analyzer determined electron kinetic energy, using pass energy of 150 V for wide survey scans and 50 V for high-resolution scans. Samples were ion-etched using 4 kV Ar ions, which were rastered over an area of 2.25 × 4 mm with total ion beam current of 2 mA, to remove adventitious carbon. Spectra were referenced to adventitious C 1s at 284.5 eV. CasaXPS software was used for XPS data analysis with Shirley backgrounds. The lithium and NCM cathode samples were lightly washed in pure diglyme before XPS measurements. Also, the samples were transferred in an airtight Argon filled puck from the glove box to the XPS chamber. Hence, there is minimal or no exposure to air.

**Floating-point experiment**. Floating-point experiments were performed in a cell comprising of lithium vs. NCM using various electrolytes reported in the main text. The batteries were charged at constant current of 0.4 mA cm$^{-2}$ up to different voltages from 3.6 to 4.3 V and then held at a constant voltage for 24 h and the values of the leak current at various voltages were measured.

**Fourier transform infrared spectroscopy**. The NCM electrodes were harvested after constant voltage charge at 3.8 V for 24 h in a battery comprising of lithium anode and NCM cathode using the electrolyte of diglyme–LiNO$_3$–HFiP with and without LiBOB additive. After drying for 24 h in the glove-box antechamber ATR-FTIR was used in the wavelength range of 800 and 4000 cm$^{-1}$.

**3-Electrode voltammetry**. 3-Electrode cell was prepared in a vial-type cell that comprised of an Ag/AgCl electrode (prior soaked in standard 1 M KCl brine) as the reference electrode and stainless steel disc (2 mm) as the working electrode at room temperature. The scan rate utilized was 10 mV/s.

**Battery performance**. 2032 type Li||stainless-steel coin cells with and without HFiP additive in diglyme–LiNO$_3$ electrolyte were prepared inside an argon-filled glove box. The amount of electrolyte used for all battery testing was 60 μl. The cells were evaluated using galvanostatic cycling in a Neware CT-3008 battery tester. CE test was performed in Li||stainless steel cell with different current densities with one each cycle comprising of 1 h. Half-cell test was performed in Li||NCM at different C-rates after initial two formation cycles of C/10. The cathode loading was 2 mAh/cm$^2$ and all the Li||NCM experiments were performed using a thin lithium (50 μm) as anode. Unless stated in the figure, the voltage ranges were chosen to 4.2 V to 3 V. All the coin-cells were crimped to a pressure of ~2500 psi. Except for the results using the crosslinked nanoparticles electrolytes, all the battery comprised of a 3501 Celgard separator. Unless stated otherwise, the LiNO$_3$ content in the battery was r = 0.5 (molar ratio between EO

and Li ions). In all the battery measurements with the HFiP chain-transfer-agent, the amount added in the electrolyte was 1 wt%. In the measurements using LiBOB salt additive, the amount utilized was 0.4 M.

## Data availability

All datasets generated and analyzed during the current study are available from the corresponding authors (L.A.A. and J.L.M.-C.) on reasonable request.

## Code availability

All computer codes used for data analysis are available from the corresponding authors (L.A.A. and J.L.M.-C.) on reasonable request.

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

## Acknowledgements

The work was supported by the National Science Foundation, Division of Materials Research, through Award No. DMR-1609125. S.C., S.S. and Q.Z. acknowledge partial support from the Beijing Institute of Collaboratory Innovation (BICI). J.L.M.-C. and A.

N. thank the High-Performance Computer cluster at the Research Computing Center (RCC) in Florida State University (FSU) for providing computational resources and support. J.L.M.-C. gratefully acknowledges the support from the Energy and Materials Initiative at FSU. A portion of this work was performed at the National High Magnetic Field Laboratory, which is supported by National Science Foundation Cooperative Agreement No. DMR-1644779 and the State of Florida. M.J.Z. and L.F.K. acknowledge support by the NSF (DMR-1654596). This work made use of the Cornell Center for Materials Research (CCMR) Shared Facilities with funding from the NSF MRSEC program (DMR-1719875). Additional support for the FIB/SEM cryo-stage and transfer system was provided by the Kavli Institute at Cornell and the Energy Materials Center at Cornell, DOE EFRC BES (DE-SC0001086).

## Author contributions

S.C., Z.T., A.N., J.L.M.-C. and L.A.A. conceived the idea and designed the experiments. S.C., Z.T., S.S. and D.V. performed the battery measurements. A.N. and J.L.M.-C. performed the simulations. S.C., Y.D., Q.Z., D.V. and S.S. performed the electrochemical characterizations. M.J.Z. and L.F.K. did the cryo-electron microscopy. All authors contributed in writing the paper.
