## [Peer Review File · Nature Communications]

Reviewers' comments:

Reviewer #1 (Remarks to the Author):

This paper reported how to stabilize the Li/NMC battery with diglymes solvent-based electrolyte. In the first part, the authors present an increasing Coulombic efficiency and cycling stability of Li metal anode with the additive of Tris(hexafluoro-isopropyl) phosphate (HFIP). In the second part, the authors report the Lithion polymer coating on cathode can broaden the electrochemical window of diglymes solvent to satisfy the NMC cathode. And in the third part, they show another additive LiBOB can also enable the NMC cathode good cycling stability in this electrolyte. This is an interesting paper, and bring new ideas to the battery community. This paper can be published in Nature Communications after addressing questions below.

1, The HFIP additive can stabilize the diglymes with Li metal anode, the author claim HFIP can quench the anionic polymerization of glyme molecules by the chain transfer mechanism. At least one more example should be given to prove this fundamental mechanism.

2, The Coulombic efficiency of Li plating and stripping can be increased to 98%, indicating that the electrolyte is still not stable enough for this battery system and it is still not high enough for a full battery. In addition, it is hard to determine whether the high stability of lithium metal anode comes from the stable polymerization by CTA, or just from a stable SEI such as the LiF and other inorganic compounds.

3, Many literatures have reported that the LiNO₃ salts can help to form a stable SEI on Li metal anode, the authors should give more discussion on it. Moreover, LiNO₃ is known for high hygroscopicity.

4, The addition of LiBOB into the electrolyte can increase the stability of NMC cathode. Does it stabilize the polymer or the CEI on the cathode? Many published literatures have reported and confirmed the LiBOB can form a stable SEI on cathodes.

5, In the final fabricated cell, the electrolyte used HFIP to stabilize the Li metal anode, used LiBOB to stabilize the NMC cathode. However, there are also papers reported the HFIP can produce stable CEI on the cathode (Journal of The Electrochemical Society, 160 (2) A285-A292 (2013) and LiBOB is useful for a stable SEI on Li metal anode. it is too simple to classify function of additives used here.

6, If all the proposed mechanism is correct, is this mechanism applicable to other lithium salts such as LiTFSI?

7, The XPS spectra are not well calibrated. LiF in F 1s should be at 685.2 eV, not 684 as discussed in the main text. The most suitable calibration is LiF F 1s for battery electrodes with LiF signal. It is suggested to calibrate all the XPS spectra.

In the experimental section, the authors state that 'Spectra were referenced to adventitious C 1s at 284.5 eV. CasaXPS software was used for XPS data analysis with Shelby backgrounds.' First, C 1s is not well calibrated to 284.5 for C-C signal, in Figure S9 and S17. Second, as far as this reviewer knows, there's no 'Shelby background' in CasaXPS. Usually, it's Shirley type. However, it seems linear ones were applied in most of the XPS spectra.

Response to Reviewer Comments

Reviewer #1 (Remarks to the Author):

This paper reported how to stabilize the Li/NMC battery with diglymes solvent-based electrolyte. In the first part, the authors present an increasing Coulombic efficiency and cycling stability of Li metal anode with the additive of Tris(hexafluoro-isopropyl) phosphate (HFIP). In the second part, the authors report the Lithion polymer coating on cathode can broaden the electrochemical window of diglymes solvent to satisfy the NMC cathode. And in the third part, they show another additive LiBOB can also enable the NMC cathode good cycling stability in this electrolyte. This is an interesting paper and bring new ideas to the battery community. This paper can be published in Nature Communications after addressing questions below.

Response: We thank the reviewer for the constructive comments on the manuscript.

1, The HFIP additive can stabilize the diglymes with Li metal anode, the author claim HFIP can quench the anionic polymerization of glyme molecules by the chain transfer mechanism. At least one more example should be given to prove this fundamental mechanism.

Response: There are several additional examples of molecules that can quench the anionic polymerization of molecular ethers by mechanisms similar to the specific inhibitor used in the study. Among these molecules BTFE (Bis(2,2,2-trifluoroethyl)) and metal salts containing TFSI (bis(trifluoromethanesulfonyl)imide stand out for their compatibility with ether-based electrolytes (ACS Appl. Mater. Interfaces, 2014, 6 (11), pp 8006–8010; Nano Energy, 40, 2017, pp 607-617) and for their stability in the presence of cathode and anode chemistries of contemporary interest. We have added a brief discussion of these points on pg. 9, first paragraph, of the revised manuscript.

2, The Coulombic efficiency of Li plating and stripping can be increased to 98%, indicating that the electrolyte is still not stable enough for this battery system and it is still not high enough for a full battery. In addition, it is hard to determine whether the high stability of lithium metal anode comes from the stable polymerization by CTA, or just from a stable SEI such as the LiF and other inorganic compounds.

Response: We agree that the C.E. observed at 1mA/cm² is not high enough for commercialization of lithium metal batteries based on this electrolyte. However, subsequent work in the group has already shown that simple complementary changes to the electrode design can lead to CEs as high as 99.7%, which are approaching values that would be of commercial interest. Thus while we expect that more optimization will be needed to achieve the better than 99.9% CE values required for commercial LMBs with N/P ratios approaching unity, we believe that the fresh concepts presented in the manuscript provide the strong foundation needed to move the community of practitioners towards CE requirements for commercially viable (N/P ratios < 4) LMBs.

3, Many literatures have reported that the LiNO₃ salts can help to form a stable SEI on Li metal anode, the authors should give more discussion on it. Moreover, LiNO₃ is known for high hygroscopicity.

Response: As highlighted on pages 5, 8, 9 and 11 of the revised manuscript, LiNO₃ based electrolytes were used in this study as the control for essentially the opposite reason. While the reviewer is right that LiNO₃ has been widely discussed as a salt additive that is able to passivate the Li anode, the mechanism is essentially unknown and few studies have challenged the broad assertions about LiNO₃'s effectiveness in stabilizing Li anodes with in-depth studies. Here, by using the LiNO₃ electrolyte as a baseline, we show that even at high salt concentrations, LiNO₃ is rigorously incapable of preventing the degradation of glymes during long term cycling. We show further that they are incapable of operation with high voltage cathode chemistries, including NCM or LCO. These findings are in fact what motivated us to search for underlying, fundamental approaches that could lead to the breakthroughs needed to achieve long-term stability of polyether-based electrolytes and both electrodes.

4, The addition of LiBOB into the electrolyte can increase the stability of NMC cathode. Does it stabilize the polymer or the CEI on the cathode? Many published literatures have reported and confirmed the LiBOB can form a stable SEI on cathodes.

Response: The reviewer's point is again well taken. While the reviewer is correct that several studies have indeed been reported showing the beneficial aspects of LiBOB additives in carbonate-based electrolytes from a battery-level perspective, in search of more molecular answers, our goal here is to understand the fundamental mechanism. Our findings that the supramolecular anionic aggregates formed when BOB reacts with ethers function in an analogous manner to an ion-rectifying (Lithion) cathode electrolyte interphases (CEI) deposited directly on a highly oxidizing cathode to facilitate de-solvation of the electrolyte salt is to our knowledge a first. This type of understanding is obviously important for designing CEI chemistries that outperform both the LiBOB and Lithion systems used as demonstrations of the broader concepts presented in the present work.

5, In the final fabricated cell, the electrolyte used HFiP to stabilize the Li metal anode, used LiBOB to stabilize the NMC cathode. However, there are also papers reported the HFip can produce stable CEI on the cathode (Journal of The Electrochemical Society, 160 (2) A285-A292 (2013) and LiBOB is useful for a stable SEI on Li metal anode. it is too simple to classify function of additives used here.

Response: We could not agree more with the reviewer's overall point, particularly in the present context where the highly reactive Li metal anode will degrade all electrolyte components in contact with its surface to form the SEI. This general point is confirmed further with XPS analysis of the lithium metal surface extracted from a cycled Li||NCM cell. The results presented in Supplementary Figure S17 clearly show that while BOB components do

not feature as dominantly in the composition of the SEI, they are present. We note however that the complementary experiment in which the anodic stability of electrolytes containing HFiP (but no LiBOB) were measured (see Supplementary Figure S16) clearly show that HFiP plays no obvious role in enhancing the oxidative stability of the electrolyte.

6, If all the proposed mechanism is correct, is this mechanism applicable to other lithium salts such as LiTFSI?

Response: The reviewer points out an important aspect of the generality of the proposed concept. As the proposed stabilization mechanism involves chemical reorganization of the glyme molecules at the anodic and cathodic interfaces, it is applicable to other lithium salts.

7, The XPS spectra are not well calibrated. LiF in F 1s should be at 685.2 eV, not 684 as discussed in the main text. The most suitable calibration is LiF F 1s for battery electrodes with LiF signal. It is suggested to calibrate all the XPS spectra. In the experimental section, the authors state that 'Spectra were referenced to adventitious C 1s at 284.5 eV. CasaXPS software was used for XPS data analysis with Shelby backgrounds.' First, C 1s is not well calibrated to 284.5 for C-C signal, in Figure S9 and S17. Second, as far as this reviewer knows, there's no 'Shelby background' in CasaXPS. Usually, it's Shirley type. However, it seems linear ones were applied in most of the XPS spectra.

Response: We apologize for the error in calibration and the typo in the methods section. We have corrected both in the revised manuscript.

REVIEWERS' COMMENTS:

Reviewer #3 (Remarks to the Author):

Most of the concerns have been addressed well in the response letter, and it can be accepted by Nature Communications. However, there is still one minor issue need to be clarified. Author claimed in the response to question 2: "subsequent work in the group has already shown that simple complementary changes to the electrode design can lead to CEs as high as 99.7%". Could author show more details and discussions on this, and add it into the manuscript?

Response to Reviewer Comments

REVIEWERS' COMMENTS:

Reviewer #3 (Remarks to the Author):

Most of the concerns have been addressed well in the response letter, and it can be accepted by Nature Communications. However, there is still one minor issue need to be clarified. Author claimed in the response to question 2: “subsequent work in the group has already shown that simple complementary changes to the electrode design can lead to CEs as high as 99.7%’. Could author show more details and discussions on this, and add it into the manuscript?

Response: We have added additional discussions in the revised manuscript.